# Highly efficient gene transfer in the mouse gut microbiota is enabled by the IncI$_2$ conjugative plasmid TP114

Kevin Neil[1], Nancy Allard[1], Frédéric Grenier[1], Vincent Burrus [1] & Sébastien Rodrigue [1✉]

The gut microbiota is a suspected hotspot for bacterial conjugation due to its high density and diversity of microorganisms. However, the contribution of different conjugative plasmid families to horizontal gene transfer in this environment remains poorly characterized. Here, we systematically quantified the transfer rates in the mouse intestinal tract for 13 conjugative plasmids encompassing 10 major incompatibility groups. The vast majority of these plasmids were unable to perform conjugation in situ or only reached relatively low transfer rates. Surprisingly, IncI$_2$ conjugative plasmid TP114 was identified as a proficient DNA delivery system in this environment, with the ability to transfer to virtually 100% of the probed recipient bacteria. We also show that a type IV pilus present in I-complex conjugative plasmids plays a crucial role for the transfer of TP114 in the mouse intestinal microbiota, most likely by contributing to mating pair stabilization. These results provide new insights on the mobility of genes in the gut microbiota and highlights TP114 as a very efficient DNA delivery system of interest for microbiome editing tools.

[1] Département de biologie, Université de Sherbrooke, Sherbrooke, QC J1K 2R1, Canada. ✉email: Sebastien.Rodrigue@USherbrooke.ca

Horizontal gene transfer (HGT) is a major driver of bacterial evolution, allowing several microbial species to exchange genes and adapt to various conditions[1,2]. One of the most important HGT mechanism is bacterial conjugation, a natural process during which DNA is transferred through a type IV secretion system (T4SS) from a donor to a recipient bacterium in close contact[3]. This phenomenon has been thoroughly studied in vitro using several conjugative elements[4–6]. However, relatively few studies have investigated the requirements for bacterial conjugation in natural and clinically relevant environments such as the gut microbiota[7–16]. The intestinal tract contains a wide diversity of microbial species that varies throughout the digestive system[17,18]. Understanding bacterial conjugation in the gut could provide valuable insights on the dissemination of antibiotic resistance genes and the emergence of multi-drug resistant pathogens[19].

Conjugative plasmids have been shown to transfer between different bacterial species residing in the gut microbiota[7,8] but only a few groups have quantified transfer rates in situ[9–12,20]. Many factors such as NaCl, propionate or butyrate concentrations, the presence of epithelial cells or inflammation have been shown to affect DNA transfer of specific conjugative plasmids in the gut[8,13–15]. However, these reports only address a single plasmid at a time and used different mouse models (gnotobiotic C3H[9,20], streptomycin treated BALB/C[10,13] or C57 BL/6[12]) and bacterial hosts (E. coli[13], Salmonella enterica[12], Lactococcus lactis[10], Enterococcus faecalis[9,20]), making comparisons difficult. Furthermore, some of these studies pre-mixed donor and recipient strains before introduction in mice[12,13]. As conjugation is a relatively fast process, this procedure could introduce artefacts or important biases in the quantification of conjugation rates. A standardized and robust in situ conjugation mouse model is therefore needed to systematically study the contribution of different plasmid families to the dissemination of antibiotic resistance genes in the microbiota, as well as the mechanisms required for bacterial conjugation in situ.

In this study, we investigated 13 plasmids transferring within the Enterobacteriaceae group, a phylum infamous for their rapid accumulation of resistance genes to many antibiotics currently on the market[21,22]. We established a robust mouse model to quantify DNA transfer rates in the gut microbiota and evaluated the contribution of 13 different conjugative plasmids to horizontal gene transfer in this environment. These experiments identified TP114 as a highly effective machinery for conjugative transfer in situ. Transposon insertion mutagenesis allowed the identification of an accessory type IV pilus shared among members of the I-complex family of conjugative plasmids as an essential component for TP114 conjugation in the gut. Taken together, our results suggest that mating pair stabilization is a key mechanism for conjugative transfer in the gut microbiota.

## Results

**In vitro transfer rates quantification.** A total of 13 conjugative plasmids isolated from enteric bacteria were selected to represent a large phylogenetic diversity spanning 10 incompatibility (Inc) groups[23]. All plasmids were sequenced, annotated and their resistance profiles were assessed (Supplementary Table 1). Three derivatives of Escherichia coli Nissle 1917 (EcN) with different antibiotic resistance profiles were next generated as standardized hosts for the conjugative plasmids. EcN derivatives were all resistant to streptomycin but additionally displayed resistance to either spectinomycin (KN01), chloramphenicol (KN02) or tetracycline (KN03) to facilitate donor and recipient strain discrimination in conjugation assays (Supplementary Fig. 1)[24]. Each conjugative plasmid was introduced into EcN KN01, which provides a standardized donor strain and limits host related conjugation efficiency bias for all conjugation assays. Transfer rates of all conjugative plasmids were then quantified on agar solid support (Fig. 1a) or in broth (Fig. 1b) by mixing an equal amount of EcN KN01 donor and EcN KN02 or EcN KN03 recipient bacteria before incubating for 2 h at 37 °C. Transconjugant formation frequencies of the 13 conjugative plasmids varied over 5 orders of magnitude and were in good agreement with published observations, usually with higher transfer rates on solid support[4,25–27].

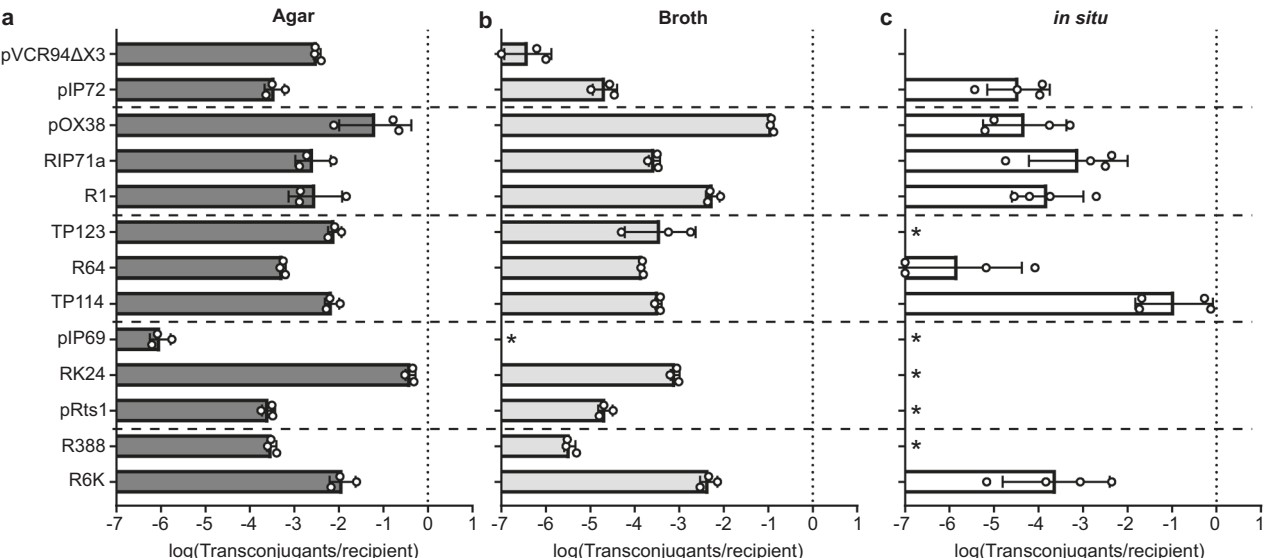

**Fig. 1 Comparison of the conjugation rates of 13 plasmids under different conditions.** A set of 13 conjugative plasmids were tested for their ability to transfer between two EcN cells during 2 h on agar (n = 3, biologically independent samples) (**a**), in broth (n = 3, biologically independent samples) (**b**), and for 24 h in situ (n = 4, biologically independent mice) (**c**) in a murine gut conjugation model. The dotted vertical line highlights a conjugation rate that allows all recipient cells to receive a plasmid. Asterisks denote the absence of detectable transconjugant colonies. White circles show individual values for each dataset. Bar graphs represent the average and error bars the standard deviation of the data set.

**Establishment of a mouse model**. A standardized mouse model was next developed to quantify the transfer rates of conjugative plasmids in the gut microbiota. C57 BL/6 female mice were selected as they were previously used for in situ conjugation assays and offer the possibility to access many transgenic strains. Mice were provided with varying concentrations of streptomycin in drinking water to deplete endogenous populations of enterobacteria in their digestive tract and to facilitate the establishment of streptomycin-resistant EcN derivatives (Supplementary Fig. 2a-f)[28]. The addition of 1.0 g/L of streptomycin in mice drinking water reliably increased EcN levels by >100-fold in most sections of the gut, as well as in feces (Supplementary Fig. 2g). This step was found to be important to achieve reproducible colonization levels by EcN, which are in turn crucial to enable the detection of low-frequency conjugative transfer events. While EcN was detected throughout the entire intestinal tract, colonization levels were higher from the caecum to the colon, with particularly high occurrences in feces for streptomycin treated mice (Supplementary Fig. 2g). Stool sampling can thus be used to facilitate manipulation, reduce the number of mice required per experiment, and maximize the sensitivity of conjugation assays.

**Quantification of in situ DNA transfer rates**. In situ transfer rates of the 13 selected conjugative plasmids were next evaluated in the murine gut using 2 consecutive gavages. The EcN recipient strain (KN02 or KN03) was first administered before allowing mice to rest for 2 h. A second gavage was then performed with the donor strain EcN KN01 carrying one of the 13 conjugative plasmids. Conjugation events were quantified by counting colony forming units (CFU) per gram of feces collected daily for three days after introduction of the EcN KN01 donor strain (Fig. 1c and Supplementary Fig. 3). No detectable levels of in situ transfer were observed for 6 out of 13 conjugative plasmids. The remaining 7 plasmids showed transfer rates ranging over ~5 orders of magnitude (Fig. 1c). No correlation was observed between in situ transfer rates and those measured on agar ($R^2 = 0.0687$) or in broth ($R^2 = 0.2727$). However, we noticed that the frequency of transconjugants detected at day 4 in the caecum were generally consistent with those obtained from feces ($R^2 = 0.701$), suggesting that stool sampling provided an appropriate proxy for in situ quantification of conjugation (Fig. 2a). A few exceptions were noted, for example with R6K, R1, and pOX38 that produced a relatively low number of transconjugants exclusively or predominantly in the caecum. Given that the density of E. coli per mg of material is similar between feces and the caecum of streptomycin-treated mice (Supplementary Fig. 2g), this phenomenon was likely due to the greater sensitivity of the assay in

the caecum from which ~10-fold more material can be analyzed, hence facilitating the detection of low frequency transfer events.

**Highly effective transfer of TP114 in situ**. Of all tested plasmids, IncI$_2$ plasmid TP114 clearly displayed the highest transfer rate in situ, showing dissemination to a very large proportion of probed EcN recipient bacteria within 24 h (Fig. 1c), which later virtually reached 100% (Fig. 2a, b). Conjugation of TP114 also showed very similar efficiencies between feces and the caecum (Fig. 2a). A few parameters of the mouse model were next modified to investigate their potential impact on TP114 transfer rates. The effect of introducing the donor bacterium EcN KN01 12 h (rather than 2 h) after the introduction of the recipient strain was first evaluated, without any clear difference on conjugation rates of TP114 (Fig. 2b) or colonization levels of EcN KN02 (Fig. 2c). The impact of the streptomycin treatment on TP114 transfer rates was also investigated, but no significant difference was observed (P value 0.1681) despite lower colonization levels by the donor and recipient bacteria (Fig. 2d). Taken together, these experiments demonstrated the robustness and reproducibility of plasmid TP114 high transfer rates in situ.

**High-density transposon mutagenesis (HDTM) of TP114**. Deciphering the molecular mechanisms used by the highly efficient plasmid TP114 could provide valuable insights on the strategies used by conjugative plasmids to disseminate within the gut microbiota. Using a high-density transposon mutagenesis (HDTM) approach, TP114 mutant libraries comprising on average one transposon insertion every ~5 bp were generated (Supplementary Table 2). The HDTM library was first analyzed in EcN, in the absence of any recipient strain, to identify the key elements for TP114 maintenance. In this context, transposon insertion sites were found to be uniformly distributed across the TP114 sequence, sparing only the loci encoding replication, maintenance or selection markers such as the replication initiation protein (repA) and the kanamycin resistance gene (aph-III) (Fig. 3a yellow boxes track 1). A conjugation assay on agar solid support was then performed between the initial HDTM library and the EcN KN02 recipient strain (Fig. 3a, track 2 and Supplementary Fig. 4a). The resulting transconjugants were also used for an additional round of transfer on agar support (Fig. 3a, track 3 and Supplementary Fig. 4a). HDTM biological triplicates were found to be highly correlated (Supplementary Fig. 4b). Gene essentiality after conjugative transfer was investigated using a ratio of the normalized transposon insertion read counts of transconjugants compared to the initial mutant library (Supplementary Fig. 4c–f). As expected, the T4SS and relaxosome

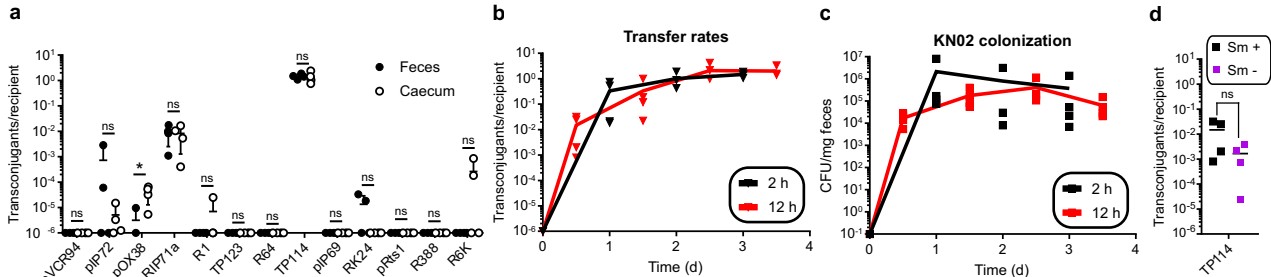

**Fig. 2 Evaluation of mouse model parameters for in situ conjugation. a**, Comparison of the transfer rates observed in the caecum and feces 4 days after the introduction of the donor strain harboring each of the 13 conjugative plasmids ($n = 4$, biologically independent mice). Error bars span the minimum and maximum observed. Effect of time between gavages on TP114 transfer rates ($n = 4$, biologically independent mice) (**b**), as well as on recipient colonization level ($n = 4$, biologically independent mice) (**c**). **d**, Conjugation rates after 12 h in mice treated or not with 1000 mg/L of streptomycin ($n = 4$, biologically independent mice). Panels **a** and **d**, One-way ANOVA statistical analysis results are shown as: ns $P > 0.05$, *$P < 0.05$. **a–d**, lines represent the average of the data.

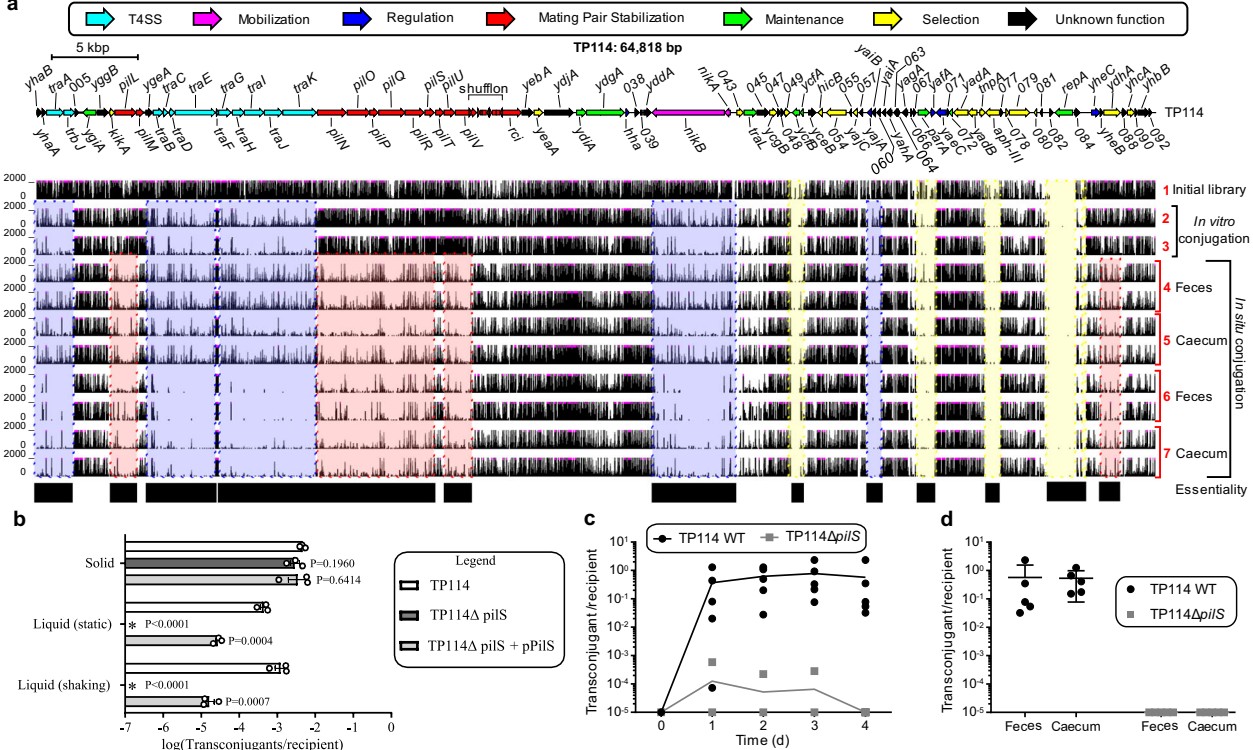

**Fig. 3 Mating Pair Stabilization (MPS) is essential for *in situ* conjugation. a** TP114 annotated map (top panel) and transposon insertions sites (lower panel) obtained from different high-density transposon mutagenesis (HDTM) mutant libraries. Gene essentiality is highlighted in yellow boxes for plasmid maintenance, blue boxes for in vitro transfer and red boxes for in situ transfer and summarized in the "Essentiality" track (track 1 to 3 $n = 3$, biologically independent samples, track 4 to 7 $n = 6$, biologically independent mice). **b** The role of the type IV pilus major pilin subunit gene (*pilS*) in MPS was confirmed in vitro by conjugation in stable (solid) and unstable (liquid) conditions ($n = 3$, biologically independent samples). Error bars show standard deviation of the mean from at least 3 biological replicates, which are shown on graph as white circles. **c** TP114Δ*pilS* mutant in situ transfer was followed for 4 days in feces then (**d**) compared with efficiencies from the caecum ($n = 5$, biologically independent mice).

encoding genes, which were already known to be essential for transfer in conjugative plasmids, were also deemed essential for TP114 to perform conjugation in vitro (Fig. 3a blue boxes track 2 and 3).

**Identification of essential genes for in situ conjugation of TP114**. TP114 HDTM libraries were also used for the evaluation of TP114 transfer in the mouse intestinal tract. The initial library maintained in the EcN strain, as well as the transconjugant library hosted in EcN KN02 obtained after a single in vitro transfer, were used as donors for in situ transfer assays as described above (Supplementary Fig. 4a). The resulting transconjugants were isolated from feces (Fig. 3a red boxes track 4 and 6) or from the caecum (Fig. 3a red boxes track 5 and 7), and their transposon insertion sites were sequenced. Interestingly, in situ conjugation required genes predicted to encode a Type IV Pilus (T4P) in addition to genes that were essential for transfer on agar solid support. T4P generally comprise a machinery responsible for the secretion of a major and a minor pilin, which were previously reported to be involved in Mating Pair Stabilization (MPS)[29]. The role of the T4P was directly tested in TP114 using a deletion mutant of the predicted major pilin *pilS*. While no difference in conjugative rates was observed when using a solid support, TP114Δ*pilS* transfer was completely abolished in broth, a phenotype that could be partially complemented using a PilS expressing plasmid (Fig. 3b)[29]. Using the *pilS* mutant for in situ conjugation yielded >1000-fold fewer transconjugants (Fig. 3c), with transfer rates ($\leq 10^{-4}$) consistent between the feces and the caecum (Fig. 3d). These results confirmed the critical role of the

T4P for in situ conjugation of TP114 and suggest that MPS is crucial for conjugation in the mouse gut microbiota.

**Comparative genomics in IncI plasmids**. Comparative genomics was next performed to investigate if the ability of TP114 to transfer at high rates in the gut microbiota was likely to be shared among other members of the IncI family. Most genes encoded by TP114 (62 out of 92), including those encoding the T4P (10 out of 10), displayed high homology (>99% similarity in nucleic acids and in protein sequences) with 7 selected plasmids of the IncI$_2$ family (Supplementary Fig. 5a, b). TP114 genes were also categorized into core, soft-core or accessory gene groups based on their distribution across IncI$_2$ plasmids (Supplementary Data 1). Comparing gene conservation and essentiality in TP114 revealed that 36 out of the 41 genes required for in situ conjugation were also part of the core genome of IncI$_2$ plasmids (Supplementary Fig. 6a, Supplementary Data 1), supporting the idea that most IncI$_2$ plasmids are likely to disseminate at high rates in the intestinal tract.

In contrast, TP114 shared very little sequence homology with plasmids of the IncI$_1$ group although these plasmids also encode a T4P. Sequence similarity between IncI$_1$ and IncI$_2$ was in fact limited almost exclusively to the *oriV* region and *pilV* gene, which encodes the minor pilin subunit (Supplementary Fig. 5c, d). However, while conjugation of the IncI$_1$ plasmid R64 did not reach high levels (average of $1.54 \times 10^{-6}$ transconjugants/ recipient) in our model, IncI$_1$ plasmid p2 was shown to transfer at high rates in situ from *Salmonella typhimurium* to *E. coli*[12,15]. Both R64 and p2 were originally isolated from *Salmonella sp.*

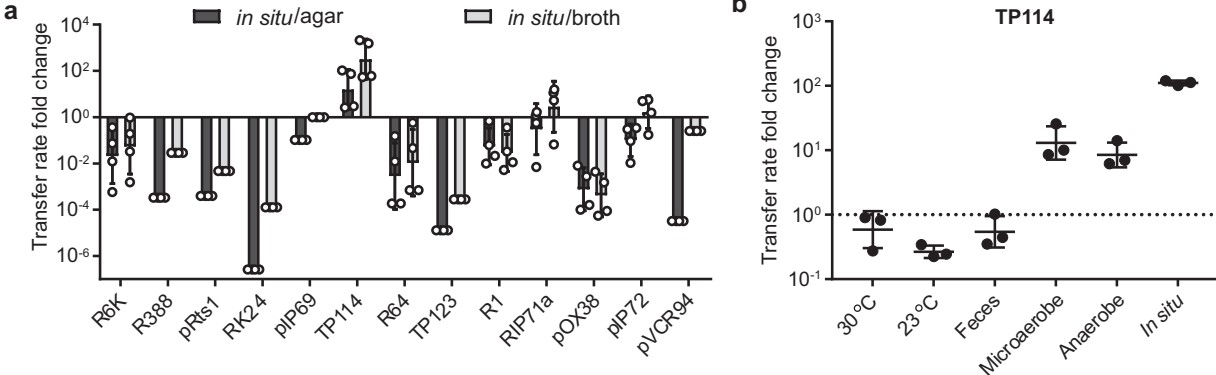

**Fig. 4 TP114 transfer is activated under hypoxic conditions. a**, ratio between the transfer rate of the 13 conjugative plasmids in situ after 24 h ($n = 4$, biologically independent mice) and their transfer efficiency on agar ($n = 3$, biologically independent samples) (dark gray) or in broth (light gray) ($n = 3$). Since R388, pRts1, RK24, pIP69, TP123, and pVCR94 did not produce any transconjugant in situ, their transfer rate fold change was calculated using the experimental limit of detection. Individual data points are shown as white circles on graph. **b**, influence of the temperature, presence of feces and hypoxic environment on the transfer rates of TP114 in relation to transfer rates for 2 h at 37 °C on agar ($n = 3$, biologically independent samples). **a** and **b**, error bars represent the standard deviation of each dataset.

raising the possibility that host cell compatibility could influence conjugation.

**Activation of TP114 under anaerobic conditions.** Of all 13 tested conjugative plasmids, TP114 presented a striking example of higher in situ conjugation relative to in vitro conditions (Fig. 3a), which led to the hypothesis that TP114 was activated by environmental conditions found in the intestinal tract. We sought to identify the potential signal that could lead to increased conjugation in the mouse intestinal tract. Conjugation assays were thus performed at different temperatures (23 °C and 30 °C), in presence of mouse feces, and under hypoxic conditions and normalized to transfer rates obtained on LB agar at 37 °C (Fig. 4b). Of these, hypoxic conditions improved the transfer rates of TP114 by ~10-fold, suggesting that TP114 responds to low oxygen concentrations found in the intestinal tract. These results also indicate that the overall performance of conjugative plasmids in the gut depends, not only on MPS, but also on appropriate regulatory responses to activate plasmid transfer in the gut.

## Discussion

Conjugative plasmids play a major role in the dissemination of antibiotic resistance. However, few studies have systematically investigated the capacity of different families of conjugative plasmid to transfer within the gut microbiota, where they could contribute to the emergence of multi-drug resistant pathogens. Using a standardized mouse model for in situ conjugation assays, our study identified IncI$_2$ plasmid TP114 as a highly effective conjugation machinery. In addition, transposon mutagenesis revealed that a T4P was essential for this function, most likely by stabilizing the interaction between the donor and recipient bacteria. With the exception of TP114, only six other plasmids (R64, pIP72, RIP71a, R1, pOX38, and R6K) were able to transfer in our in situ conjugation assays. Both IncI$_1$ plasmid R64 and IncB plasmid pIP72 encode T4P albeit without any significant sequence homology with TP114. Conjugative plasmids RIP71a, R1, as well as pOX38 all belong to the IncF group and rely on TraN, an adhesin expressed at the surface of the donor host cell, to stabilize the mating pair by binding to OmpA or LPS[30]. While no gene involved in MPS has been identified yet in R6K, this plasmid was reported to transfer with almost the same efficiency on agar or in broth[31] (Fig. 1a, b). Since MPS is required for efficient conjugation in broth, it is likely that R6K has the capacity

to stabilize the interaction between its host and a recipient bacterium although the exact genes responsible for this function remain to be identified. In fact, all plasmids that transferred in situ were also able to yield transconjugants in broth with relatively high efficiencies relative to agar solid support ($3.7 \pm 4.0$-fold lower transfer rates in broth relative to solid support). Conversely, conjugative plasmids that did not generate transconjugants in situ transferred at lower rates in broth relative to solid support ($156.7 \pm 10.6$-fold lower transfer rates in broth relative to solid support) suggesting that they could not perform MPS in the context of our assay. A clear example is IncP1α plasmid RK24, which had the highest transfer rate in vitro on agar support. However, RK24 lacks MPS genes[32], transfers ~500-fold less efficiently in broth relative to agar support, and had virtually no conjugation activity in situ. These observations further support the importance of MPS for plasmid conjugation in the intestinal microbiota.

Plasmids that failed to produce transconjugants in our in situ assays could possibly encode genes that bear no homology to known MPS systems, and use them only in other contexts. For example, their adhesins may recognize structures found on other bacterial species that are not displayed on the EcN recipients. MPS would hence play a very important role in determining plasmid transfer range towards specific species or strains in certain unstable environments. In silico analyses were performed to identify putative MPS genes in the six conjugative plasmids that had no detectable transfer in the mouse intestinal tract, and no putative adhesin-related protein domains could be found in RK24, R388, or pIP69. However, possible orthologs of TraN were identified in pVCR94, pRts1, and TP123. The protein sequence of these orthologs were found to diverge from the TraN present in IncF plasmids, suggesting that their role or specificity could differ (Supplementary Fig. 6b). It is also possible that the TraN variants of pVCR94, pRts1, and TP123 interact more strongly with other bacterial species that are potentially more closely related to their original hosts (Supplementary Table 1) or simply lack the capacity to perform MPS. Another factor that might influence conjugation rates is the availability of commensal recipient bacteria that could sequester broad-host-range plasmids and limit their ability to be transferred to our introduced EcN recipient strain. However, the transfer range of most conjugative plasmids is still largely unknown, making this hypothesis difficult to assert at this point.

The ability of conjugative plasmids to transfer in situ involves MPS but also requires other specialized machineries such as the

T4SS and the relaxosome. However, using computational analyses to associate each plasmid to a specific incompatibility (Inc; replication initiation protein (RepA)), mobilization machinery (MOB; relaxase), or mating pore formation (MPF; T4SS VirB4 subunit) group showed that diverse plasmid phylogenetic families can support conjugative transfer in the mouse intestinal tract (Supplementary Table 1). This observation implies that in situ conjugation is not restricted to a specific type of T4SS or plasmid family.

Conjugation in the gut microbiota was initially thought to occur in a relatively stable matrix akin to an agar plate[16]. This conclusion was based on experiments with IncF plasmid R1 in which transfer frequencies were compared in situ and in vitro in broth or on agar. Based on these results, it was suggested that bacterial conjugation occurs most frequently in the mucus layer of the intestinal lumen after biofilm formation[16]. However, several factors can influence conjugation rates in situ. We identified MPS as an important molecular mechanism for conjugation in the gut at early stages of colonization. This suggests that DNA transfer occurs in a rather unstable environment. The mouse model used in this study most likely represents a situation in which an incoming bacterium is introduced in the gut microbiota and can disseminate plasmids in the microbiota. Plasmids that were unable to transfer in situ in our assays could potentially transfer at later stages of colonization, for instance when embedded in dense biofilms. RK24 is an interesting example as this plasmid was previously used as a mobilization machinery for microbiome editing, and was shown to produce interspecies transconjugants[7,33]. However, conjugation rates were not reported[7] or shown to be very low[33], which is consistent with our results.

IncI$_2$ plasmids likely share the ability to transfer DNA at high frequencies in the gut. Surveillance of plasmids from this family could provide insights on the likeliness of antibiotic resistance gene dissemination in animals and humans. As several plasmids are often found in a single host[34], investigating the interactions between different conjugative plasmid families in situ could provide further insights on the mobility of resistance genes. For example, it is possible that mobile genetic elements unable to perform MPS in situ could take advantage of the MPS mechanism of other plasmids to transfer more efficiently in the gut. If so, interfering with MPS could become an interesting strategy to prevent dissemination of antibiotic resistance in infected patients and animal husbandry.

Given that the microbiota plays major roles in human health, different strategies to correct imbalanced microbiome compositions have been recently proposed. For example, microbiota-editing strategies could rely on bacteriophages or conjugative plasmids to disseminate CRISPR or toxin genes to kill specific bacteria[33,35–37]. However, these studies often highlight the lack of efficient conjugative plasmid for DNA delivery[33,35]. Highly effective conjugative systems such as TP114 could thus become important DNA delivery tools in the development of microbiome editing technologies.

## Methods

**Strains, plasmids, and growth conditions**. Strains and plasmids are described in Supplementary Table 3 and are available upon request. Cells were typically grown in Luria broth Miller (LB) or on LB agar medium supplemented, when needed, with antibiotics at the following working concentrations: ampicillin 100 μg/mL, chloramphenicol 34 μg/mL, kanamycin 50 μg/mL, nalidixic acid 4 μg/mL, rifampicin 66.7 μg/mL spectinomycin 100 μg/mL, Sm 50 μg/mL, sulfamethoxazole 160 μg/mL, tetracycline 15 μg/mL, and trimethoprim 32 μg/mL. Diaminopimelic acid (DAP) auxotrophy was complemented by adding DAP at a final concentration of 57 μg/mL in the medium. All cultures were routinely grown at 37 °C. Cells with thermosensitive plasmids (pSIM6, pE-FLP, pGRG36, pFG036) were grown at 30 °C. Bacterial cultures were grown for no more than 18 h.

**DNA manipulations**. A detailed list of oligonucleotide sequences is found in Supplementary Table 3. Cloning vectors were prepared using EZ10-Spin Column Plasmid Miniprep kit (Bio Basic) whereas genomic DNA (gDNA) and conjugative plasmids were extracted using the Quick-gDNA miniprep (Zymo Research) according to the manufacturer's instructions. PCR amplifications were performed using Veraseq DNA polymerase (Enzymatics) or TaqB (Enzymatics) for DNA parts amplification and screening, respectively. Digestion with restriction enzymes (NEB) were incubated for 1 h at 37 °C following manufacturer's recommendations. Plasmids were constructed by Gibson assembly[38] using the NEBuilder Gibson Assembly mix (NEB) following manufacturer's protocol. These procedures were applied to the construction of pPilS that are further detailed in the Supplementary methods section.

**DNA purification**. Purification of DNA was performed between each step of plasmid assembly to prevent buffer incompatibility or interference with enzymatic reactions. For general purposes, DNA was purified by Solid Phase Reversible Immobilization (SPRI) using HiPrep PCR DNA binding beads (MagBio). For sequencing, DNA was purified using Agencourt Ampure DNA binding beads (Beckman Coulter). Protocols were identical for both types of beads and followed manufacturer's guidelines. For restriction enzyme digestions, DNA was purified using DNA Clean and Concentrator (Zymo Research) following manufacturer's recommendation to prevent buffer incompatibility between the digestion's CutSmart buffer (NEB) and the polyethylene glycol-based buffer of the SPRI beads, which greatly decreased DNA yield during purification. After purification, DNA concentration and purity were routinely assessed using a NanoDrop spectrophotometer (Thermo Fisher Scientific).

**DNA transformation into *E. coli* by electroporation**. Routine plasmid transformations were performed by electroporation. Electrocompetent *E. coli* strains were prepared by sub-culturing in 20 mL of LB broth. Cultures reaching an optical density of 0.6 at 600 nanometers (OD$_{600nm}$) were washed three times in sterile distilled water. Cells were then resuspended in 200 μL of water and distributed in 40 μL aliquots. The DNA was then added to the electrocompetent cells and the mixture was transferred in a 1 mm electroporation cuvette. Cells were electroporated with a pulse of 1.8 kV, 25 μF, and 200 Ω for 50 ms. Cells were then resuspended in 1 mL of non-selective LB medium to recover for 1 h at 37 °C or 30 °C (for thermosensitive plasmids) before plating on selective media.

**DNA transformation into *E. coli* by heat-shock**. Heat-shock transformation was mostly used for Gibson assembly products transformation. Chemically competent cells were prepared according to the rubidium chloride protocol as described previously[39]. Chemically competent cells were flash-frozen and conserved at −80 °C before use. Routinely, up to 10 μL of DNA was added to 100 μL of EC100Dpir+ chemically competent cells before transformation with a 45 s heat-shock at 42 °C. Cells were then resuspended in 1 mL of non-selective LB medium to recover for 1 h at 30 °C or 37 °C before plating on selective media.

**Recombineering**. All recombineering experiments were performed using pSIM6 as described previously[40,41]. Briefly, the recombineering cassette was electroporated in an induced *E. coli* + pSIM6 strain. Then, cells were left to recover overnight at room temperature before plating on selective medium. Single isolated colonies were stabbed in a new plate before homogenization in 100 μL of 5% w/v Chelex solution (Bio-Rad). Afterwards, the Chelex mixture was heated to 56 °C for 25 min and 100 °C for 10 min in a PCR machine for DNA extraction. For PCR screening, 1 μL of freshly extracted DNA was added to the PCR mix. Positive clones were sub-cultured in 5 mL of selective LB broth overnight and frozen in 25% glycerol for storage. This protocol was applied to the deletion of the *pilS* in TP114 and the deletion of *dapA* in the chromosome of EcN, which are further described in the Supplementary methods section.

**Generation of modified EcN strains**. The modified EcN strains were generated by Tn7 insertion of the antibiotic resistance cassettes as described previously[24] with minor adaptations. More specifically, the pGRG36 vector was purified from *E. coli* EC100Dpir+ and digested using SmaI and XhoI. The inserts were amplified by PCR using their corresponding primers (Supplementary Table 3) and cloned by Gibson assembly (NEBuilder Gibson Assembly kit) between attL$_{Tn7}$ and attR$_{Tn7}$ sites of the digested pGRG36 plasmid (Supplementary Fig. 1a–c) following manufacturer's recommendations. The Gibson assembly products were then transformed in a chemically competent *E. coli* EC100Dpir+. Integrity of the constructions was confirmed by restriction enzyme digestions and positive clones were transformed into *E. coli* MFDpir+[42]. Plasmid pGRG36 derivatives were then mobilized towards EcN by the RK24 conjugative machinery of the MFDpir+ strain. To mediate cassette insertion into the terminator of *glmS*, EcN was first grown at 30 °C in LB with 1% arabinose until 0.6 OD$_{600nm}$. Then, cells were heat-shocked at 42 °C for 1 h and incubated at 37 °C overnight to allow for plasmid clearance. Cells were then streaked onto an LB agar plate selecting only the insert. Plasmid elimination was tested by streaking ≥20 colonies on both plates with or without ampicillin, which selected for the pGRG36 backbone. Colonies that only grew in absence of ampicillin but contained the insert selection markers were then

confirmed for insertion using the corresponding primers from Supplementary Table 3.

**In vitro conjugation assays**. All in vitro conjugation assays used derivatives of KN01ΔdapA as the donor strain and KN02 or KN03 as the recipient strain depending on the antibiotic markers carried by the plasmid tested. Both strains were sub-cultured from frozen stocks up to 18 h prior to conjugation experiments and were then mixed at a 1:1 volume ratio (100 μL each), centrifuged at $20,000 \times g$ for 1 min and washed in 200 μL of LB without antibiotics. The cell mix was then centrifuged again and either resuspended in 5 μL of LB broth and deposited on a LB agar plate with DAP or resuspended to 1.0 $OD_{600nm}$ in LB broth with DAP. The cell mixture was then incubated at 37 °C for 2 h unless specified otherwise. Conjugation experiments in shaking broths were performed on a tube tumbler rotating mixer (VWR, #SBS550-2) at 20 rpm while conjugation in static broth were performed in microtubes on a tube-holder. After incubation, cells were resuspended in sterile PBS, diluted 1/10 serially and 5 μL of each dilution were spotted in triplicates on LB plate with antibiotics selecting donors, recipients, or transconjugants. All conjugation frequencies were reported as a factor of the recipient CFUs. The conjugation frequencies calculated on a "per donor" basis were equivalent since cells were mixed 1:1. For experiments involving variation in temperature during conjugation, the mix of donor and recipient strains were incubated either at 23 °C, 30 °C, or 37 °C for 2 h before CFU assessment. For experiments with feces in the conjugative mix, feces were obtained from healthy C57 BL/6 mice, which received no treatments. Feces were then homogenized following procedures described in the Feces sampling section of the Supplementary Methods section. Then, feces were transferred directly in the conjugation mix and incubated for 2 h at 37 °C before plating for CFU. No colonies that could have been erroneously interpreted as transconjugants were detected when feces were plated on selective medium containing appropriate antibiotics. Conjugation under hypoxic condition were carried using Gaspack anaerobe (BD 260001) for anaerobic condition and Gaspack $CO_2$ (BD 260679) for microaerobic condition. Conjugation were incubated in an airtight bag for 24 h to allow for the creation of hypoxic conditions. All conjugation experiments were repeated in at least three independent biological replicates.

**Mouse model**. All mice-related protocols followed the *Université de Sherbrooke Animal Care Comity Guidelines* and were strictly evaluated to avoid animal suffering. Animals were provided with water and standard chow (Charles River) *ad libitum* during the experiments. Animals were housed in individually ventilated cages and no more than 5 individuals shared the same cage. All animals used were C57 BL/6 females of 16–20 g (Charles River) and were given ≥3-days rest upon arrival. Animal weight and health was evaluated daily, and no notable health or weight loss was noted for all mice throughout the experiments. For Sm-treated mice, 1000 mg/L of Sm was added to drinking water 2 days prior to gavages. From that point, water bottles were changed every 3 days to maintain Sm efficiency. The bacterial strains were prepared as described in the mice inoculum preparation section of the Supplementary Methods section. Briefly, for each time a bacterial strain was introduced in mice, approximately $1 \times 10^8$ viable cells (quantified by CFU) in PBS were administered by gavage to each mouse using a curved 1.5-inch 20 g ball-tip needle. Bacterial presence was monitored in feces using CFU counts at specified time points during the experiments. Further details on the preparation of the inoculums needed for mice gavages, as well as feces sampling and mice dissection procedures are available in the Supplementary Methods section.

**In situ conjugation assays**. For in situ conjugation experiments, mice received the recipient strain 2 or 12 h prior to the introduction of donor strain to avoid possible plasmid transfer in the PBS solution prior to gavage. Conjugation was then monitored in feces samples at specified time points. Mice were sacrificed at the end of the experiment and the caecum was extracted to measure conjugation levels in the murine gut. Feces were homogenized and CFUs were counted on MacConkey plates as described in the *Feces sampling* section.

**DNA sequencing**. Illumina sequencing of conjugative plasmids and of HDTM mutants was performed at the *Plateforme RNomics* (https://rnomics.med.usherbrooke.ca/) of the *Université de Sherbrooke*. Oxford Nanopore sequencing was also performed on TP114 to help resolve regions difficult to assemble such as the shufflon. In addition, Sanger sequencing was performed by the *Plateforme de séquençage et de phénotypage* (http://www.sequences.crchul.ulaval.ca/) of the *Université Laval* on TP114 to confirm its sequence and fill gaps. Specific details on the sequencing and assembly of conjugative plasmids and of the HDTM experiments can be found in the Supplementary Methods section.

**Analysis of plasmid gene function and generation of phylogenetic trees**. In silico analysis of plasmid gene function was performed using both CDsearch[43] and BLASTp v2.8.1[44]. A protein multi-fasta file was first generated for all Open Reading Frames (ORF) predicted by RAST[45]. Using CDsearch, TP114 genes with known protein domains or superfamilies were attributed functions. For genes in which CDsearch failed to identify any protein domain with high confidence (e-value < 1 × $10^{-15}$), their corresponding protein sequence file was also submitted to BLAST v2.8.1 to identify putative protein homologs. Analyses were performed using

default parameters. BLAST hits with high identity levels (>98% identity on >95% query coverage) were used to attribute putative functions only when more than five hits showed the same result. Proteins that failed to match these criteria were considered of unknown function. The protein homologs of VirB4, the relaxase and the replication initiation protein were identified following the same procedures described above for all 13 conjugative plasmids and were next aligned with each other using ClustalW and inputted in MEGA-X v10.0.5[46] for the generation of neighbor-joining phylogenetic trees with 1000 bootstrapping replicates.

**High-density transposon mutagenesis**. A conjugation-assisted random transposon mutagenesis experiment was performed. The transposition system was composed of pFG036 (a plasmid coding for a cI transcription repressor) and pFG051 (a Pi-dependent suicide plasmid coding for the Tn5 transposon machinery under the repression of cI, a RK24-based origin of transfer and a spectinomycin resistance transposon) cloned into *E. coli* MFDpir+[42]. This DAP-auxotrophic strain contains an RK24 conjugative machinery integrated in its genome and expresses the Pi protein required for pFG051 replication. The HDTM experiment was performed in several successive steps to clearly identify the function of genes involved at each one of these steps. First, pFG051 was transferred by conjugation from MFDpir+ to EcN containing TP114 for 2 h at 30 °C on LB + DAP plates in triplicates. Once in EcN, the Tn5 machinery was constitutively expressed from pFG051 to mediate random transposon insertions in TP114. Then, transconjugants were entirely spread onto 6 plates per replicate and incubated overnight at 37 °C to form a cell lawn. After the incubation, cells were collected using a cell scrapper and subsequently resuspended in LB broth with selective antibiotics. A 100 μL aliquot of the transconjugants was used in two subsequent conjugative transfer experiments performed in vitro and in situ using KN02 and next KN03 as recipients. For each of these experiments, transconjugants forming the mutant library were washed, resuspended in 4.5 mL of LB + 25% glycerol and cryopreserved prior to DNA extraction and sequencing.

**Mouse model for in situ HDTM library conjugation**. A cryopreserved 500 μL aliquot of the HDTM mutant library was inoculated in 20 mL selective LB broth and incubated at 37 °C for 4 h before gavage. The recipient strains were prepared as described in the in situ conjugation section. Briefly, frozen stocks were streaked on MacConkey for overnight growth before sub-culturing in 5 mL LB. Then, the recipient strains were sub-cultured again in 20 mL LB 2 h prior to gavage. When ready, cells were washed once in PBS and concentrated in a volume equivalent to 6.0 $OD_{600nm}$. Mice were administered the recipient strain 3 h prior to the introduction of the donor strain. Conjugation was then monitored after feces sampling at 24 and 48 h. At 48 h, mice were sacrificed, and the caecum was extracted. At each time-point, $4 \times 100$ μL of sample per mice were also plated to maximize the number of transconjugants for sequencing.

**HDTM libraries analysis**. Reads were first trimmed based on their quality and the presence of the Nextera Illumina adapter using Trimmomatic v0.32[47] with the parameters SLIDINGWINDOW:4:20 and MINLEN:30. The quality of the reads, before and after trimming, was assessed with FastQC v0.11.4 using the default parameters[48]. Reads mapping on the EcN chromosome were filtered out and the remaining reads were mapped onto TP114. These alignments were done with BWA MEM v0.7.2 using the default parameters[49]. Alignments with a mapping quality score lower than 30 were discarded. The position of the middle base pair of the 9-bp Tn5 insertion site duplication was then used to represent every corresponding alignment[50]. Insertion sites represented by only one read were discarded to filter out sequencing noise. The insertion maps with normalized read counts (based on samples sequencing depths) were then visualized using UCSC Genome Browser in a Box v1.12[51]. The essentiality of the genes in condition 1 (initial library) was verified manually, searching for low coverage regions that were mappable and reproducible in all three replicates. A read count table was then generated by calculating the normalized read counts per gene for each condition. Insertion sites in the first 5% and last 15% of the gene were discarded from the read count as they may lead to functional gene fragments. The genes important for in vitro and in situ conjugation were determined based on the normalized gene read count ratio between condition 1 and the test condition in the following manner: (Normalized Read count × – Normalized Read count 1)/Normalized Read count 1. A core set of genes visually identified in UCSC Genome Browser as essential for conjugation in vitro (*traABCDEGHIJK*, *trbJ*, *nikAB*) and in situ (*pilLNOPQRSUV*) was then used to set the maximal ratio value for each condition. All genes with gene count ratios bellow the maximal value were considered essential in the given condition.

**Comparative genomics using BRIGG**. Gene content comparison was performed on TP114 against a database of 7 randomly selected plasmids of the $IncI_1$ and $IncI_2$ subfamilies. The BRIGG stand-alone software[52] was used to perform BLAST-based homology analysis between TP114 and each plasmid group. Homology was analyzed using both the nucleotide sequence of the whole plasmids and amino-acid sequences of the coding genes. Conservation of genes was evaluated using the sequence identity cut-offs of 100%, 70%, and 50%. The identity percentage was calculated by attributing scores of −2 for mismatches, +1 for matches and a linear cost for insertion/deletion. Genes were then categorized as core genes when present

in 100% of the plasmids, soft core genes when present in more than 50% of the plasmids, or accessory genes when present in less than 50% of the plasmids.

**Statistics and reproducibility**. Statistical significance was performed on the logarithmic value of the data using One-way ANOVA unless specified otherwise. $P$-values are directly indicated on the graphs and represent statistical significance of the difference between the two data groups. Differences in the data were considered significant when the $P$-value was below 0.05. When $P$-value are not noted directly on graph, statistical significance was noted as: ns $P > 0.05$, *$P < 0.05$, **$P < 0.01$, ***$P < 0.001$. Number of replicates is specified in "$n =$" statements in figure legends. All in vitro experiments were performed in biological triplicates using three independently grown cultures. For experiments involving mice, a minimum of 4 C57 BL/6 female mice were used for each sampling.

**Reporting summary**. Further information on research design is available in the Nature Research Reporting Summary linked to this article.

## Data availability

The sequence of conjugative plasmids is available in GenBank (see Supplementary Table 1 for accession numbers). Transposon insertion sites for the HDTM experiments (Fig. 2) are available on UCSC Genome Browser under http://genome.ucsc.edu/cgi-bin/hgTracks?genome=TP114&hubUrl=https://datahub-103-cu2.p.genap.ca/Neil_2020/hub.txt. Reads for the sequencing of TP114, other conjugative plasmids and HDTM experiments are available at NCBI under BioProjects: PRJNA510561, PRJNA576482, and PRJNA510811 respectively. Source data for the main figures is available in Supplementary Data 2.

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

## Acknowledgements

We thank Dominick Matteau for assistance in the sequencing of TP114, the Plateforme Rnomics of Université de Sherbrooke for assistance with Illumina sequencing, as well as Compute Canada and Compute Quebec for access to bioinformatics resources and support. We are grateful to Pr. Laura Frost for the kind gift of pOX38. We also acknowledge all lab members for thoughtful discussions. This work was supported by the Canadian Institutes of Health Research (CIHR #159817). S.R. holds a Chercheur boursier junior 2 fellowship from the Fonds de recherche du Québec–Santé (FRQS), and K.N. is the recipient of a graduate research scholarship form the Fond de Recherche du Québec–Nature et Technologies (FRQNT) and from the Natural Science and Engineering Research Council of Canada (NSERC). N.A. is supported by a doctoral scholarship from the Université de Sherbrooke.

## Author contributions

K.N. and S.R. designed the experiments; K.N. and N.A. performed the experiments; K.N. analyzed the data; F.G. performed bioinformatic analyses for the HDTM and conjugative plasmids sequence assembly except for TP114, which was performed by K.N.; S.R. supervised the project. K.N. and S.R. wrote the manuscript; N.A., V.B., and F.G. revised the manuscript.

## Competing interests

The work presented in the manuscript is part of US provisional patent application 62/696,367. All of the authors of the present manuscript except for F.G. are also co-authors of this provisional patent application. K.N. and S.R. have a financial interest in TATUM bioscience. F.G. declares no competing financial interests and all authors declare no competing non-financial interests.
