## [Peer Review File · Communications Biology]

Reviewers' comments:

Reviewer #1 (Remarks to the Author):

Review comments on the manuscript by Neil et al.

In their manuscript, Neil and colleagues have studied plasmid conjugation in situ in mice. The study is well-designed and clearly significant effort has been made to find out details behind the observed conjugation patterns. They show that plasmid TP114 is very efficient in dispersing itself in situ. A notable transposon mutant library was constructed for this plasmid in order to track down specific genetic qualities that may be responsible for the high conjugation rate. Indeed, they show that mating pair stabilization appears to be crucial for conjugation to occur at least in certain conditions. Several other plasmids were also able to transfer in mice gut, probably owing to their mating pair stabilization qualities. The authors do a great job at discussing potential explanations why some of the plasmids may be unable to transfer in situ regardless of their efficient transfer on solid support. Indeed, bacteria often harbor multiple plasmids and it may be necessary only for one of the plasmids to stabilize the contact with potential recipients. Also commensal bacteria and plasmid-plasmid competition may play a role. This manuscript is well-written and it does improve our understanding of plasmid dispersal in the gut (which is important for multitude of reasons, including antibiotic resistance). This reviewer would like the authors to add a sentence on how the bacteria were actually administered to the mice. At this point the manuscript appears to be very complete and as such I would recommend its publication as it is.

Reviewer #2 (Remarks to the Author):

Neil et al describe the characterization of different conjugative plasmids for transfer of DNA between donor and recipient bacteria in a mouse model. The authors identified an IncI2 conjugative plasmid that had superior conjugation transfer levels in situ compared to previous studies. This work will facilitate efforts to develop strategies to transfer genetic material into the gut microbiota for various applications. Overall, the studies are well designed, the interpretations of the results and conclusions drawn are reasonable. As such, this reviewer has enthusiasm for publication of this work in this journal.

Some suggestions that the authors can address:

1). Lines 82-84: "However, we noticed that the frequency of transconjugants detected at day four in the caecum were highly consistent with those obtained from feces ($R^2=0.701$), suggesting that stool sampling provided an appropriate proxy for in situ quantification of conjugation (Figure 2b)." seems like an over-interpretation there, some plasmids like RK24 , R6K, R1, and RIP72 have either fecal transfer or caecum transfer but not the other. The R^2 correlation is not particularly meaning analysis there. Can the authors explain why some transfer are seen in caecum or feces and not the other? The authors may want to temper down the claim that no transfer is observed in some of these.

2) The study could better highlight the technical differences between this study and previous studies which may lead to divergent results. For example, the use of EcN as recipient and different donor strain may affect in vivo viability and stability for transfer. These could better be discussed and highlighted in the discussion.

3) Genes in conjugative plasmids that may be expressed only in vivo and not in vitro could contribute to transfer differences or plasmid stability differences in the recipient may be an alternative hypothesis to discuss or test. For instance, can the authors rule out that the various plasmids are equally stable in in vitro and in vivo settings? It is possible that the transfer rates are

similar but that some plasmids such as RK24, R64, or TP123 are less stable in in vivo settings. One possible way to test this is to give these transconjugants to the mice and see how stable they are in vivo to separate out the effect of plasmid stability and transfer rates.

Reviewer #3 (Remarks to the Author):

In the manuscript presented by Rodrigue and colleagues, the authors have tested the conjugative capability of plasmids of different groups among E coli. The results are compelling and the study is very interesting. The study is novel and of interest to those in the field. My biggest concern is the mouse study though. It is not clear exactly how many mice were used for each assay and whether variations in gender was taken into account. How many replicates were performed for the mouse study?

Response to Reviewers:

We have carefully studied the comments made by the reviewers on our recent manuscript submission entitled: “**Incl₂ conjugative plasmid TP114 disseminates at very high rates in the intestinal microbiota**”. We have addressed their remarks in a revised manuscript, which we hope will now be acceptable for publication. The reviewers’ comments are shown below in black while our response is in blue.

Comments from the editors and reviewers:

-Reviewer 1

Review comments on the manuscript by Neil *et al.*

In their manuscript, Neil and colleagues have studied plasmid conjugation *in situ* in mice. The study is well-designed and clearly significant effort has been made to find out details behind the observed conjugation patterns. They show that plasmid TP114 is very efficient in dispersing itself *in situ*. A notable transposon mutant library was constructed for this plasmid in order to track down specific genetic qualities that may be responsible for the high conjugation rate. Indeed, they show that mating pair stabilization appears to be crucial for conjugation to occur at least in certain conditions. Several other plasmids were also able to transfer in mice gut, probably owing to their mating pair stabilization qualities. The authors do a great job at discussing potential explanations why some of the plasmids may be unable to transfer *in situ* regardless of their efficient transfer on solid support. Indeed, bacteria often harbor multiple plasmids and it may be necessary only for one of the plasmids to stabilize the contact with potential recipients. Also commensal bacteria and plasmid-plasmid competition may play a role. This manuscript is well-written and it does improve our understanding of plasmid dispersal in the gut (which is important for multitude of reasons, including antibiotic resistance). This reviewer would like the authors to add a sentence on how the bacteria were actually administered to the mice. At this point the manuscript appears to be very complete and as such I would recommend its publication as it is.

We would like to thank this reviewer for his/her very positive comments. The information about the method used for bacterial administration to mice was mostly presented in the supplementary methods. We understand that the administration procedure could go unnoticed in the supplementary document. We therefore included additional details in the method section at lines 371-374.

-Reviewer 2

Reviewer #2 (Remarks to the Author):

Neil *et al* describe the characterization of different conjugative plasmids for transfer of DNA between donor and recipient bacteria in a mouse model. The authors identified an Incl₂ conjugative plasmid that had superior conjugation transfer levels *in situ* compared to previous studies. This work will facilitate

efforts to develop strategies to transfer genetic material into the gut microbiota for various applications. Overall, the studies are well designed, the interpretations of the results and conclusions drawn are reasonable. As such, this reviewer has enthusiasm for publication of this work in this journal.

We would first like to thank this reviewer for the insightful comments and inputs.

Some suggestions that the authors can address:

1). Lines 82-84: “However, we noticed that the frequency of transconjugants detected at day four in the caecum were highly consistent with those obtained from feces ($R^2=0.701$), suggesting that stool sampling provided an appropriate proxy for *in situ* quantification of conjugation (Figure 2b).” seems like an over-interpretation there, some plasmids like RK24, R6K, R1, and RIP72 have either fecal transfer or caecum transfer but not the other. The R^2 correlation is not particularly meaningful analysis there. Can the authors explain why some transfer are seen in caecum or feces and not the other? The authors may want to temper down the claim that no transfer is observed in some of these.

This section was modified to better explain why we estimate that feces provide an appropriate proxy to quantify conjugation in the gut (lines 94-102). In fact, most of the work previously published on *in situ* conjugation only analyzed feces to determine transfer rates. We also make it more explicit (lines 82-84) that our strategy was designed to minimize the number of mice per experiment by using feces to quantify conjugation rates on day 1, 2, and 3 before sacrificing mice to investigate the content of the caecum only at the last day of the experiment.

The reviewer noted that in some cases, conjugative transfer was observed in the caecum but not in feces or vice versa. Since most plasmids displayed very low transfer rates (near detection limits), quantifying conjugation in the caecum likely improved the detection capacity since more material (~10x) could be retrieved from the caecum than from feces for CFU assessment (see lines 97-102). For RK24, some transconjugants were found only in feces, and not in the caecum. It is possible that, as water gets absorbed in the colon, feces could provide a more stable matrix that allowed rare conjugation events to happen without the need for mating pair stabilization. However, as the spatial distribution of conjugation events was not investigated, this hypothesis should be tested in future work.

2) The study could better highlight the technical differences between this study and previous studies which may lead to divergent results. For example, the use of EcN as recipient and different donor strain may affect *in vivo* viability and stability for transfer. These could better be discussed and highlighted in the discussion.

We added three sentences to better describe previous mouse models of bacterial conjugation at lines 37-40. Briefly, most studies investigated conjugative transfer for a single plasmid, and in some cases in a non-quantitative manner. Other studies pre-mixed the donor and recipient bacteria prior to their introduction in mouse, a practice that could result in artefacts or biases given that conjugation could occur before the gavage is performed (see lines 41-42). We estimate that this modification to the introduction will allow the reader to understand how *in situ* conjugation has previously been investigated and to better appreciate the rationale supporting our mouse model.

3) Genes in conjugative plasmids that may be expressed only *in vivo* and not *in vitro* could contribute to transfer differences or plasmid stability differences in the recipient may be an alternative hypothesis to discuss or test. For instance, can the authors rule out that the various plasmids are equally stable in *in vitro* and *in vivo* settings? It is possible that the transfer rates are similar but that some plasmids such as RK24, R64, or TP123 are less stable in *in vivo* settings. One possible way to test this is to give these transconjugants to the mice and see how stable they are *in vivo* to separate out the effect of plasmid stability and transfer rates.

Gene expression regulation is certainly an important factor contributing to *in situ* transfer rate differences between conjugative plasmids. As exemplified by TP114, some plasmid can activate the expression of genes involved in DNA transfer in response to environmental conditions experienced in the gut (lines 170-179). However, investigating plasmid regulation in the intestinal microbiota would require a substantial amount of work that is outside of the scope of the current manuscript. We are in fact currently investigating gene expression regulation in TP114 and hope to submit a manuscript on this topic in the next few months.

We have not directly quantified the stability of each plasmid through time in a formal experiment. However, Supplementary Fig. 3 reports the CFU counts for donor, recipient, and transconjugant bacteria. Since donor cells and transconjugants are conceptually very similar (a transconjugant becomes a new host for a conjugative plasmid and eventually a proficient donor cell), we expect the stability of the conjugative plasmids to be similar between these two populations. Interestingly, the abundance of donor cells remained relatively stable throughout the experiment, and we thus evaluate that plasmid stability is not a major factor limiting the observed transfer rates. Given that the donor strain tends to gradually decrease with time (even for “empty” donors cells, see Supplementary Fig. 2), and that the exact abundance in each mouse would vary in each test group, we consider that precisely investigating plasmid stability represents a considerable amount of work that expands beyond the scope of the present manuscript.

-Reviewer 3

Reviewer #3 (Remarks to the Author):

In the manuscript presented by Rodrigue and colleagues, the authors have tested the conjugative capability of plasmids of different groups among *E. coli*. The results are compelling and the study is very interesting. The study is novel and of interest to those in the field. My biggest concern is the mouse study though. It is not clear exactly how many mice were used for each assay and whether variations in gender was taken into account. How many replicates were performed for the mouse study?

The sex of the mice used in our experiments is reported in the methods section at lines 366-368, and additional details are now provided in the results section at lines 72-74. Only females C57/BL6 were used in our experiments as they are more docile than males and were used in previously published studies quantifying conjugation *in situ* (all previous models presented in the introduction section, ref 9-12, 20). The difference in sex was therefore not directly accounted for in this study, but we do not expect the results to be significantly different using male mice as the diet and microbiome composition should be similar.

Most experiments were performed with four mice per test group (the HDTM experiment was performed with two mice per replicate per experiment (6 mice per experiment)). We added the number of mice used per group in the figure legends and in statistics and reproducibility section in the methods section (lines 475-476).